

# Transformation of *n*-alkanes from plant to soil: a review

Carrie L. Thomas[1,2], Boris Jansen[2], E. Emiel van Loon[2], and Guido L. B. Wiesenberg[1]

[1]Department of Geography, University of Zurich, CH-8057 Zurich, Switzerland
[2]Institute for Biodiversity and Ecosystem Dynamics, University of Amsterdam, Amsterdam, 1098XH, Netherlands

*Correspondence to*: Carrie L. Thomas (carrie.thomas@geo.uzh.ch)

**Abstract.** Despite the importance of soil organic matter (SOM) in the global carbon cycle, there remain many open questions regarding its formation and preservation. The study of individual organic compound classes that make up SOM, such as lipid biomarkers including n-alkanes, can provide insight into the cycling of bulk SOM. While studies of lipid biomarkers, particularly n-alkanes, have increased in number in the past few decades, only a limited number have focused on the

transformation of these compounds following deposition in soil archives. We performed a systematic review to consolidate the available information on plant-derived n-alkanes and their transformation from plant to soil. Our major findings were 1) a nearly ubiquitous trend of decreased total concentration of n-alkanes either with time in litterbag experiments or with depth in open plant-soil systems, 2) a decrease in either Carbon Preference Index (CPI) or Odd-over-Even Predominance (OEP) with depth, indicating degradation of the n-alkane signal or a shift in vegetation composition over time, and 3) preferential

degradation of odd chain length and shorter chain length n-alkanes. The review also highlighted a lack of data transparency and standardization across studies of lipid biomarkers, making analysis and synthesis of published data time-consuming and difficult. We recommend that the community move towards more uniform and systematic reporting of biomarker data. Furthermore, as the number of studies examining the complete leaf-litter-soil continuum is very limited as well as unevenly distributed over geographical regions, climate zones, and soil types, future data collection should focus on underrepresented

areas as well as quantifying the transformation of n-alkanes through the complete continuum of plant to soil.

## 1 Introduction

Soil organic matter (SOM) is one of the largest terrestrial reservoirs in the carbon cycle, containing a carbon stock more than ten times greater than that of forest biomass (Settele et al., 2015). Despite its importance, there remain many gaps in the knowledge of the formation, degradation, and preservation of SOM (Schmidt et al., 2011). Studying the degradation and

preservation of various individual compound classes included in bulk SOM, such as lipids, could increase the overall understanding of SOM dynamics.

In the last few decades, there has been a myriad of studies using lipids in soil as molecular proxies for a variety of purposes, including paleoecology and paleoclimate reconstructions (Jansen and Wiesenberg, 2017). Many of these compounds are

considered "biological markers," or "biomarkers" in short, because they can be indicative of their source organisms and may



be preserved following deposition in environmental archives, such as soils and sediments (Peters et al., 2005). *n*-Alkanes, in particular, have been used extensively due to their potential for preservation and because their extraction and analysis are relatively easy compared to other compound classes (Diefendorf and Freimuth, 2017).

The primary source of *n*-alkanes in soils and sediments are cuticular waxes of plant leaves and roots (Eglinton and Eglinton, 2008). Plant-derived *n*-alkanes are characterized as having long chains of carbon atoms and a strong predominance of molecules with an odd number of carbons (Kolattukudy et al., 1976). Because *n*-alkanes are ubiquitous in plant leaves and roots, the distribution pattern of *n*-alkanes across multiple chain lengths is used as a proxy to determine past vegetation species, not merely the presence of individual *n*-alkanes with specific chain lengths (Jansen et al., 2006). Therefore, the preservation

of these patterns following deposition in an archive is essential to their continued use as a proxy for paleovegetation, mediated by (and therefore also an estimator of) contemporary climatic and hydrological conditions. (e.g., Bush and McInerny, 2015).

To aid in characterizing *n*-alkane patterns, various index measurements have been developed over the years. These include the Carbon Preference Index (CPI), Odd-over-Even Predominance (OEP), and Average Chain Length (ACL). The CPI and OEP

were both developed initially as a method for identifying sources of petroleum (Bray and Evans, 1961; Scalan and Smith, 1970). They have since been used extensively in studies in other environmental settings such as soil and sediments to determine the sources and degree of degradation of *n*-alkanes (e.g., Zhou et al., 2010; Trigui et al., 2019). Higher values for the CPI and OEP indicate *n*-alkanes derived from plant waxes and are also characteristic of well-preserved biomarker signals (Marzi et al., 1993; Hoefs et al., 2002). Lower values are typical of *n*-alkanes originating from other sources such as microbes or indicate a

high degree of degradation (Marzi et al., 1993; Hoefs et al., 2002). The ACL is a weighted average of the chain length distribution of the odd-chain lengths of *n*-alkanes. It has been used to differentiate between dominant vegetation types, such as woody plants versus grasses, and as an indicator of environmental conditions such as drought (e.g., Crausbay et al., 2014; Wüthrich et al., 2017).

The equations used for each of these indices often vary between studies, particularly in terms of what chain lengths are included. Basic equations for CPI (Marzi et al., 1993), OEP (Hoefs et al., 2002), and ACL (Poynter et al., 1989) are as follows:

CPI: $\frac{(\sum_{i=n}^{m} C_{2i+1}) + (\sum_{i=n+1}^{m+1} C_{2i+1})}{2(\sum_{i=n+1}^{m+1} C_{2i})}$ (1)

where n is the starting *n*-alkane divided by 2 and m is the ending *n*-alkane divided by 2.


OEP: $\frac{(C_{27} + C_{29} + C_{31} + C_{33})}{(C_{26} + C_{28} + C_{30} + C_{32})}$ (2)

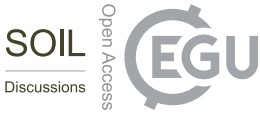

$$ACL: \frac{(\sum C_i \times i)}{\sum C_i} \tag{3}$$

where $C_i$ is the concentration of the *n*-alkane containing *i* carbon atoms.


Though there have been many studies regarding the diagenesis of *n*-alkanes and other biomarkers in sediments (e.g., Cranwell, 1981), only a few have focused on the potential degradation or transformation of these molecules in soil (Jansen and Wiesenberg, 2017). This review aims to consolidate the available information on the fate of *n*-alkanes in soil and how their distribution patterns may be altered from fresh plant material to litter to soil. Clearer knowledge of degradation and

transformation will lead to a better understanding of the accuracy of biomarker observations and more careful interpretation; it can also lay the basis for a better process understanding of degradation and transformation. Ultimately, this understanding can be translated into observation-models to standardize biomarker observations and render them more useful.

## 2 Methods

The review was performed systematically. A Boolean search string was developed to search all of the databases, except the

Derwent Innovation Index, included in the Web of Science: (("leaf wax*" OR "lipid biomarker*" OR "alkane*" OR "n-alkane*" OR "chemical fossil*" OR "epicuticular wax*" OR "molecular prox*") AND ("soil* OR "peat*" OR "topsoil*" OR "litter*")). Selection criteria were also developed to guide the process of screening the search results. These include:

1. Peer-reviewed source
2. Study includes primary data or observations on the degradation of the distribution patterns, concentration, or index measurements (e.g., CPI, OEP, or ACL) of *n*-alkanes
3. Study occurs in natural soil or peat with minimal contamination OR in lab conditions if microbial degradation of *n*-alkanes in natural soils was investigated
4. No enhanced or augmented measures were taken to promote degradation
5. Study includes data on modern soils

The search string returned 9297 results as of August 27, 2020. Using the five selection criteria, the titles and abstracts of the search results were screened for suitability, and 8911 results could be rejected without examining the full text. Seven records were not available in English, and ten records were duplicates.


From 37 studies, the following types of information were extracted if available:

1. Molecular data (*n*-alkane concentration, CPI, OEP, ACL)
2. Environmental data (elevation, mean annual air temperature (MAAT), mean annual precipitation (MAP))
3. Soil data (soil type, pH, total organic carbon (TOC))
4. Vegetation data (Species, general vegetation)

Studies including data obtained using pyrolysis were not included as this method provides indirect measurements of *n*-alkanes.



For studies that included concentration data but not index measurements, Equation 1 was used to calculate CPI with n = 11 and m = 16, Equation 2 was used to calculate OEP, and Equation 3 was used to calculate ACL using odd chain lengths between 27 and 33.

Figures 1-6 were created with the "ggplot2" (Wickham, 2016) and "ggpubr" (Kassambara, 2020) packages, and Figure 7 was created with packages "sp" (Pebesma et al., 2005; Bivand et al., 2013), "sf" (Pebesma, 2018), "tmap" (Tennekes, 2018), and "rnaturalearth" (South, 2017) using R version 3.6.2 (R Core Team, 2019). Full species names of vegetation are provided where available. Complete data can be found in the supplementary material.

## 3 Results

### 3.1 *n*-Alkanes in the litter layer

Prior to incorporation in soil, plant debris can form a litter layer on top of the soil, especially in forest ecosystems. This plant debris is the principal source of material for the formation of soil organic matter in the topsoil (Kögel-Knabner, 2002), and degradation of plant-derived *n*-alkanes begins in the litter layer. There have been 27 studies examining the degradation of *n*-alkanes in the litter, including six litterbag experiments as well as studies assessing the litter layer in open plant-soil systems, which we define as natural systems including no manipulation.

### 3.1.1 Results from litterbag experiments

We found a total of six litterbag experiment studies during the review ranging in time intervals from 300 days to 23 years (Table 1). The species investigated for *n*-alkane degradation included *Calluna vulgaris* in a peatland setting (Huang et al., 1997), *Acer pseudoplatanus*, *Fagus sylvatica*, and *Sorbus aucuparia* in forest settings (Zech et al., 2011; Nguyen Tu et al., 2017), *Setaria viridis, Eleusine indica, Amaranthus retroflexus* and *Erigeron speciosus* deposited at an agricultural experimental field (Wang et al., 2014), *Neosinocalanus affinis* and *Osmanthus fragrans* (Li et al., 2017). Additionally, Schulz et al. (2012) measured degradation of *Zea mays* and *Pisum sativum* in an incubation experiment using agricultural soil.

The primary trend spanning nearly all (five out of six) of the litterbag experiments is a considerable decrease in the total concentration of *n*-alkanes compared to the initial biomass (µg/g dry weight). The majority of species contained a much lower relative concentration of *n*-alkanes in the final measurement of the experiments (Fig. 1). The largest decreases in the first 100 days occurred in litter from *N. affinis* (Li et al., 2017), *Z. mays,* and *P. sativum* (Schulz et al., 2012) (Fig. 1). In contrast, *n*-alkanes in *E. speciosus* and *S. viridis* actually increased in concentration by the end of the experimental period (Wang et al., 2014) (Fig. 1). From the beginning to the end of the litterbag experiments, the distribution patterns of the *n*-alkanes remained relatively constant with the same range of chain lengths and the dominant chain length staying the same in nearly all of the species (Supplementary Material).





The ACL was reported in four of the studies; the results from Huang et al. (1997), Li et al. (2017), Wang et al. (2014), and
Nguyen Tu et al. (2017) are shown in Fig. 2a. There were only small variations in ACL over the course of the studies. The *N. affinis* litter showed the largest increase in ACL (Li et al., 2017), while *C. vulgaris* was the only species that ended the experiment with a lower ACL than the initial measurement (Fig. 2a).

The CPI was also reported in the same four studies; the results are shown in Figure 2b. There is some variability between the studies in how the CPI changed with time. The CPI of *N. affinis* dropped rapidly at the beginning of the experiment but then slightly increased until the end of the experiment (Li et al., 2017) (Fig. 2b). In contrast, Nguyen Tu et al. (2017) measured an increase in the CPI of *F. sylvatica* litter from 9.9 to 14.7 after two years (Fig. 2b). *C. vulgaris* and *O. fragrans* showed slight decreases in CPI over the experimental periods, respectively 23 years and 369 days. Three of the grasses in the experiment of
Wang et al. (2014) revealed a similar pattern with small increases or decreases at different sampling points but returning to about their starting CPI after 210 days (Fig. 2b). However, *A. retroflexus* showed a different pattern with a sharp increase at 60 days, followed by a decrease to 1.4 from its initial 3.7. Zech et al. (2011) reported OEP rather than CPI (Fig. 2b). They found that *A. pseudoplatanus* was characterized by a small initial increase and then remained about the same. Both *S. aucuparia* and *F. sylvatica* had a significant decrease in OEP over 27 months.

**3.1.2 Results from the plant-litter-soil continuum in open plant-soil systems**

Twenty-one studies compared the *n*-alkane compositions of litter to that of fresh plants or soil in open plant-soil systems without using litterbags (Fig. 3, Table 2). These include field sites from a range of climates and environmental conditions. To examine the presence of general trends throughout the data, we clustered them into six broad classes of biomes: coniferous forest, deciduous forest, mixed forest, grass- or shrubland, peatland, and steppe. The choice for these biomes was based on the
abundance and distribution of the study sites for which results were found.

The majority of sites included in these studies showed a clear decrease in absolute *n*-alkane concentration from fresh leaves to senescent leaves (if included) to litter to the organic layer (if included) to the topsoil (Fig. 3, Supplementary Material) (e.g., Chikaraishi and Naraoka, 2006; Zhang et al., 2017; Marseille et al., 1999; Otto and Simpson, 2005; Nguyen Tu et al., 2001).
From either the fresh plants or the litter to the topsoil, the total concentration of *n*-alkanes decreased significantly. The average percent decrease from litter to topsoil across the included studies was 46%, while that from fresh plants to topsoil was 87%. In Fig. 3a, the total concentration normalized to dry weight is shown along with the total concentration normalized to total organic carbon (TOC) in Fig. 3b. The primary difference between the trends shown in Figs. 3a and 3b is that for coniferous and mixed forest vegetation, the concentration of *n*-alkanes increases along the continuum when normalized to TOC.




The studies reported various changes to the distribution patterns of the *n*-alkanes from plant to litter to soil (Supplementary Material). Some of the results showed contradicting trends; for example, Zhang et al. (2017) measured an increase in long-chain *n*-alkanes ($C_{27}$-$C_{33}$) from fresh plants to litter in *Sphagnum*-dominated peatlands and a decrease in mid-chain *n*-alkanes ($C_{21}$-$C_{25}$), while Marseille et al. (1999) found evidence of increasing proportions of mid-chain lengths in deeper litter layers in
forest soils under *Fagus sylvatica*. Other studies noted decreases in the relative concentration of long-chain *n*-alkanes (Chikaraishi and Naraoka, 2006; Otto and Simpson, 2005; Hirave et al., 2020), while Nguyen Tu et al. (2001) noted a preferential decrease in shorter chain lengths from fresh leaves to litter of *Gingko biloba*.

Figures 4a, 4b, and 4c show the changes in ACL, CPI, and OEP from plant to soil that were either reported or calculated with
provided data. The ACL typically increased from fresh plant material to topsoil, though not at sites with grass- or shrubland vegetation and mixed forest vegetation (Fig. 4a). Both the CPI and OEP decreased from fresh material to topsoil in all vegetation types except for coniferous forest (Figs. 4b, 4c).

### 3.1.3 Results from soil profiles

With time, partially degraded plant material from the litter layer will be incorporated into the topsoil. Degradation will continue
in the soil, aided by soil microbiota and organisms, such as earthworms. Below the topsoil, there is a relative increase in root-derived carbon, including lipids, and a simultaneous decrease in litter-derived carbon (Angst et al., 2016). Limited studies have shown that root biomass could be an important source of *n*-alkanes in soil, particularly in the subsoil under certain environmental circumstances, such as the presence of a nutrient-rich fossil subsoil horizon (Jansen and Wiesenberg, 2017). We found a large number of studies containing *n*-alkane measurements from soil profiles and grouped these in the same six
biomes as previously mentioned.

Figure 5a shows the changes in the total concentration of *n*-alkanes normalized to dry weight in the six biomes. Figure 5b also presents concentration data but normalized to TOC. The latter was only reported for studies in the deciduous forest, grass- or shrubland, and peat vegetation types. Below the uppermost mineral soil horizon, the results varied per study and biome. In
areas with coniferous forest vegetation, there was some evidence of increasing concentration of *n*-alkanes with increasing depth within the A horizons (Marseille et al., 1999; Schäfer et al., 2016), though the majority of sites still had decreasing concentrations (Schäfer et al., 2016) (Fig. 5a). There were mixed results at the sites with deciduous forest vegetation, depending on whether the *n*-alkane concentration was normalized to the TOC (Figs. 5a, 5b). Within the A horizons, conflicting trends were identified in two studies; Anokhina et al. (2018) found an increase in *n*-alkane concentrations by a range of 66.8% to
120.9%, while Schäfer et al. (2016) noted decreases in concentrations ranging from 33.0% to 89.3%. Further down the profiles, Anokhina et al. (2018) found a significant increase in concentration in an E horizon, followed by a decrease in a B horizon. Other studies noted generally decreasing concentration through the whole profile (e.g., Angst et al., 2016; Bull et al., 2000; Cui et al., 2010; Wu et al., 2019). At sites with grassland vegetation, from the first to the second horizon, there was typically



a decrease in the concentration of *n*-alkanes (Marseille et al., 1999; Schäfer et al., 2016; Bull et al., 2000), though there were
three sites where *n*-alkanes increased in concentration by a range of 4.8% - 22.2% (Celerier et al., 2009; Schäfer et al., 2016)
(Fig. 5a). Only two studies on grasslands reported on horizons below the A: Celerier et al., 2009 found the concentration
decreased from the A to B horizon by 53.56% while Feng and Simpson (2007) noted increases in the concentration of *n*-
alkanes $C_{24}$-$C_{33}$ from the A to B horizon in four grasslands ranging from 100% to 350%, followed by further changes from the
B to C horizon ranging from a 25.0% decrease to a 314.3% increase. Another study in a steppe biome measured not by horizons
but by 15 cm intervals, and this study found alternating increases and decreases in the *n*-alkane concentrations down to a depth
of 97.5 cm (Buggle et al., 2010).

The soil contained a wider range of chain lengths than plants or litter (e.g., Angst et al., 2016; Anokhina et al., 2018), though
no study reported measuring *n*-alkanes shorter than $C_{14}$. Soils were also reported to contain more *n*-alkanes with even chain
lengths than the plants or litter, indicative of a source other than higher terrestrial plants (Almendros et al., 1996). Even so, the
dominant chain length in many studies remained the same at all depths (e.g., Buggle et al., 2010; Cui et al., 2010; Huang et
al., 1996) or was replaced by a longer chain length in the lower depths (e.g., Angst et al., 2016; Anokhina et al., 2018; Bull et
al., 2000). Every study, except one (Almendros et al., 1996), reported a dominant chain length of at least $C_{25}$ in the soil layers,
showing that the most abundant source of *n*-alkanes is leaf wax from higher plants.

As with the litter studies, not every study included the same type of index measurements. A few studies reported the ACL (Fig.
6a), which decreased with depth at many sites (e.g., Wu et al., 2019), but also increased at a significant number of sites (e.g.,
Schäfer et al., 2016). The CPI is presented in Fig. 6b and was generally found to decrease with depth in the majority of studies
that included it (e.g., Angst et al., 2016, Celerier et al., 2009; Huang et al., 1996; Wu et al., 2019). In other sites, the OEP (Fig.
6c) was found to decrease with depth at most sites (e.g., Schäfer et al., 2016). However, there were a few instances of the OEP
increasing in lower soil layers, including at the Anokhina et al. (2018) study site, in which the OEP had decreased to ~1 in the
12-28 cm interval then increased in the following 28-60 cm interval.

## 4 Discussion

Overall, the review indicated that there is a very limited number of studies that examined the degradation of *n*-alkanes either
in manipulative experiments or in open systems. The studies that are available focus on very specific settings and are modest
in scope and size, which makes it difficult to determine what the primary drivers of change are and how these might cause a
variance between environmental settings and vegetation species. Studies with available data are unevenly distributed
geographically, with the vast majority performed in Europe. Therefore, more research focusing on *n*-alkane degradation along
the entire pathway from plant to soil is urgently needed, and future research should be performed on a wider geographic range.




Despite these limitations, there are some trends emerging from the literature available to date with respect to *n*-alkane degradation along (parts of) the trajectory plant-litter-soil. These trends are as follows:

1. Decrease in total concentration of *n*-alkanes over time in litterbag experiments and with depth in open systems
2. Decrease in CPI or OEP with depth in open systems
3. Preferential degradation of odd chain lengths and shorter chain lengths

*Decrease in total concentration of n-alkanes*

There are many potential variations in the fate of *n*-alkanes in soil profiles. However, across vegetation types and study sites, there is generally a trend of decreasing overall alkane concentration with soil depth. This is most likely caused by the
degradation or reworking of *n*-alkanes by soil microbiota. Degradation products of *n*-alkanes include *n*-methyl ketones through subterminal oxidation (Klein et al., 1968; Amblès et al., 1993; Jansen and Nierop, 2009) or *n*-alcohols through terminal oxidation that can be converted into aldehydes and finally into *n*-fatty acids (Rojo, 2009). Conversely, *n*-alkanes can also be produced *in situ* through the oxidation of *n*-alkenes or *n*-alcohols or the decarboxylation of *n*-fatty acids (Jansen and Nierop, 2009).


In litterbag experiments, the total concentrations of *n*-alkanes decreased with time in nearly all of the vegetation species (Fig.1). Biodegradation is likely accelerated in the litter layer due to oxic conditions (Chikaraishi and Naraoka, 2006) and an active microbial community (Rojo, 2009). Additionally, Zech et al. (2011) acknowledged the possibility that *n*-alkanes may have been washed out of the litterbags. However, as *n*-alkanes are hydrophobic and almost insoluble in water, this is unlikely,
especially for plant-derived long-chain *n*-alkanes. Unexpectedly, each of the grass species included in the Wang et al. (2014) study showed increases in the concentration of total *n*-alkanes (Fig. 1). They explained this as the result of *n*-alkane degradation being a complex process influenced by both chemical properties and plant species.

In open plant-soil systems, the concentration of *n*-alkanes generally decreased significantly with depth, averaging an 87%
decrease from fresh plant leaves to topsoil, as seen in most of the litterbag experiments (Figs. 1, 3). However, a few of the coniferous forest sites showed higher concentrations compared to the fresh plant material in the litter or underlying horizon (Fig. 3). This is likely due to the fact that the vegetation or litter samples at these sites often had a very low initial concentration of *n*-alkanes (e.g., Schäfer et al., 2016; Otto and Simpson, 2005). Therefore, it is feasible that the accumulation of *n*-alkanes in the lower layers over time could lead to higher concentrations, even as degradation is occurring. Additionally, as waxes on
coniferous needles are renewed throughout the year, physical abrasion and subsequent deposition of the waxes onto the soil can cause an enrichment of wax lipids in the topsoil compared to the fresh biomass (Heinrich et al., 2015).

*Decrease in CPI or OEP*

With some exceptions, the CPI and OEP tended to decrease with depth or from litter to soil (Figs. 6b, 6c). In aerobic conditions,
microbial reworking has been shown to cause an increase in lower chain length *n*- alkanes and a loss of high odd-over-even

predominance (Grimalt et al., 1998; Brittingham et al., 2017). This is a result of the degradation of longer chain length *n*-alkanes occurring at the same time as an accumulation of microbial-derived medium-chain *n*-alkanes (Brittingham et al., 2017). The resulting change in chain length distribution can be the cause of the decreases in CPI and OEP with depth throughout the mineral soil that are seen in many studies (Angst et al., 2016, Celerier et al., 2009; Huang et al., 1996; Wu et al., 2019; Li et

al. 2017; Bliedtner et al., 2018; Schäfer et al., 2016). Changes in CPI could also indicate a shift in vegetation composition over time.

In the litterbag experiments, the trend was not quite as clear. For *Calluna vulgaris* and three of the grass species, only slight changes were noted (Fig. 2b). Wang et al. (2014) inferred from their results that the CPI might be insensitive to early litter

degradation, e.g., within the first year. This cannot fully explain the results of Huang et al. (1997) as the experiment lasted 23 years, however as it occurred in a peatland with anaerobic conditions present at shallow depth, it is likely that degradation was inhibited following the litterbag's burial by newer litter accumulation. Nguyen Tu et al. (2017) found that the CPI increased in *Fagus sylvatica* litter after 2 years, which can further support that there is limited degradation of the distribution pattern of *n*-alkanes in litter. Zech et al. (2011) found that there was a large decrease in OEP in *Fagus sylvatica* and *Sorbus aucuparia*,

which could indicate that there is preferential degradation of odd *n*-alkanes rather than even *n*-alkanes as the ratio approaches 1 or that there is a different source of *n*-alkanes affecting the litter's signal that has an even-over-odd predominance. Additionally, other organic compounds such as biopolymers could break down through degradation and contribute *n*-alkanes that do not have an odd-over-even predominance (Jansen and Wiesenberg, 2017).

*Preferential degradation*

As the overall distribution patterns of *n*-alkanes are often used for the purpose of vegetation reconstruction (e.g., Schwark et al., 2002), it is essential that these patterns remain relatively unchanged following deposition onto the litter layer so that they can be used accurately. All of the species in the litterbag experiments retained their range of chain lengths as well as their most abundant chain length (Supplementary Material), evincing that there are limited changes in the distribution patterns of the *n*-

alkanes and no preferential degradation of long chain lengths.

Though some of the results from the litterbag experiments are supported by those seen in the open plant-soil systems, there are also some differences. While the distribution patterns of the *n*-alkanes in the litterbag experiments were not found to change much from their original, this was not the case in all of the studies in the open systems. Differences in results from the litterbag

experiments could be as a result of the mesh bags used, preventing larger soil organisms such as earthworms from accessing the litter inside (Bradford et al., 2002).

There is a lot of evidence for microbial alterations of *n*-alkanes in open plant-soil systems and the resulting changes in distribution patterns. These changes could be partially due to preferential degradation of *n*-alkanes of certain chain lengths or

selective preservation of some alkanes (Lichtfouse et al., 1998) rather than the same rate of microbial degradation for all *n*-alkane compounds. A few studies have found that *n*-alkanes with shorter chain lengths are degraded more quickly than those with longer chain lengths (Moucawi et al., 1981; Amblès et al., 1993). Additionally, the results of the other studies included in the review appear to support this as well, considering that in the majority of the studies, the dominant long-chain *n*-alkane either remained the most abundant in the soil or was superseded by an even longer chain length (e.g., Bliedtner et al., 2018;

Buggle et al., 2010; Cui et al., 2010; Huang et al., 1996, Angst et al., 2016; Anokhina et al., 2018; Bull et al., 2000). Therefore, even if preferential degradation occurs, it is still generally possible to identify characteristic patterns of plant-derived *n*-alkanes in soil because the primary long chain lengths remain dominant.

The general decreases in CPI and OEP will not affect source apportionment or vegetation reconstruction if the odd chain length

*n*-alkanes are used. However, changes in the ACL could indicate that there has been a shift in the pattern of odd chain lengths. In terms of the biomes, the largest relatively short-term changes seem to occur the most in the deciduous forest vegetation (Fig. 6a). This could cause difficulties in identifying dominant vegetation. To determine what could be causing a change in ACL in the short-term in certain biomes, further study of preservation in specific environments should be considered. While changes in the deeper profiles could be a result of a shift in the vegetation input over time, this is not likely the case in the

studies that only considered topsoil samples, as did most of the included studies.

*Other sources of n-alkanes*

There are also other potential sources of *n*-alkanes that may affect distribution patterns and alter plant-derived signals. These include *n*-alkanes present in aerosols after being abraded from leaves (Rogge et al., 1993; Nelson et al., 2017, 2018). The leaf

waxes found in aerosols may obscure the local vegetation biomarker signal found in soil with a more regional signal (Nelson et al., 2017; Howard et al., 2018). Two additional potential sources that have been relatively neglected in studies of biomarkers are pollen (Hagenberg et al., 1990) and insects (Chikaraishi et al., 2012).

Leaves are not the only plant organs containing *n*-alkanes. Roots have been found to generally have different distributions of

*n*-alkanes than aboveground biomass (Jansen et al., 2006; Angst et al., 2016), though for grasses, the composition can be similar (e.g., Kuhn et al., 2010). Therefore, the root signal can potentially affect the distribution of *n*-alkanes in deeper soil by imprinting a new signal over that of plant waxes (Gocke et al., 2010). However, roots often contain a much lower concentration of *n*-alkanes compared to plant leaves and are not likely to be the primary source of *n*-alkanes in topsoils (Angst et al., 2016; Gamarra and Kahmen, 2015). Nevertheless, studies have found that roots in some species have a higher concentration of *n*-

alkanes than in the leaves (Huang et al., 2011) and the continuous growth of roots, as well as the release of exudates, can cause roots to be a larger source of *n*-alkanes, particularly in lower soil horizons (Jansen and Wiesenberg, 2017). Woody tissues and bark also contain low concentrations of *n*-alkanes, though they may be relevant sources in forests (Seca et al., 2000).





*Factors affecting preservation*

Additionally, there are many environmental factors that can affect the preservation of the *n*-alkane concentrations and distribution patterns in soil. These include soil characteristics. In regards to soil, the pH has been found to affect the preservation of lipids and other organic compounds, with lower rates of decomposition noted in more acidic soils (Moucawi et al., 1981; Oades, 1988; Bull et al., 2000). This is likely due to reduced microbial activity in acidic soils. Furthermore, *n*-alkanes can be physically protected from microbes due to encapsulation within larger organic macromolecules (Almendros

and González-Vila, 1987; Lichtfouse et al., 1998) or in soil aggregates (Wiesenberg et al., 2010).

*Transportation of n-alkanes in soil*

Because lipids are generally hydrophobic and not soluble in water, *n*-alkanes are not considered to be as susceptible to leaching as other organic compounds (Naafs et al., 2004). However, *n*-alkanes could be transported as particulate matter when sorbed

to minerals or other organic material. This could be a substantial factor, in particular in soil types such as Alisols and Luvisols characterized by vertical transport of clay particles (IUSS Working Group, 2014). The potential for changes in *n*-alkane concentration or composition as a result of vertical transport in soil has not yet been researched in depth.

## 5 Implications for further research

### 5.1 Potential to correct for post-depositional changes

Despite the evidence of *n*-alkane degradation, there have been very few studies that have attempted to address this when measuring and quantifying *n*-alkanes in soil archives, particularly when using the composition to reconstruct vegetation changes. A notable exception is Zech et al. (2009), who developed a two end-member model using long-chain *n*-alkane ratios from litter and topsoil under pure forest and pure grassland cover as the two end-members. Data from their study and their reviewed literature were plotted on "degradation lines," using the *n*-alkane ratios as the dependent variable and OEP as the

independent. Using these simple models, it was possible to estimate the percentage of total *n*-alkanes that had been contributed by grass species or forest species. However, Zech et al. (2009) noted that the accuracy decreased with lower OEP values as the lines converged. This method has been subsequently used in a number of studies from the same working group (Zech et al., 2012, 2013; Schäfer et al., 2016; Bliedtner et al., 2018).

Buggle et al. (2010) used a linear regression approach to calculate a relationship between long-chain *n*-alkane ratio and OEP values in a modern soil, and using this "alteration line," they corrected (i.e., removed the effect of degradation) the long-chain *n*-alkane ratios found in paleosol samples. This approach assumed that the alteration of *n*-alkanes in the paleosol would be similar to that in the modern soil. The corrected ratios were used to determine the contributions of *n*-alkanes in the paleosol that were derived from trees or grasses.


These two approaches rely on the assumption that forest and grassland vegetation can be differentiated based only on the long-chain *n*-alkane composition, which is likely not always applicable due to variations in composition across species and environmental conditions (Bush and McInerney, 2013). Therefore, there is certainly room for the development of more approaches to accounting for degradation effects on *n*-alkane compositions. However, the lack of quantitative, process-based

studies of *n*-alkane degradation in various ecosystems hampers the development of more advanced approaches to correct for *n*-alkane degradation.

## 5.2 Knowledge gaps

What this review has illustrated above all is that there are still many gaps in fully understanding the fate of *n*-alkanes in soil, particularly in quantifying post-depositional changes. This is not aided by the fact that there does not seem to be a uniform

method for reporting lipid data measured in soils. Although soil characteristics have been shown to influence the preservation of *n*-alkane composition, basic information, such as soil pH, is often not included in papers.

Additionally, there has been a noticeable shift towards data transparency and increased availability in recently published studies. However, no data and meta-data standards exist for storing and labeling biomarker data, the soil and vegetation from

which it was collected, and the lab methods used for their determination. And as a result, the biomarker data found in the literature are not interoperable (i.e., it cannot readily be processed by analysis software) (Wilkinson et al., 2016). Furthermore, the data from older but still frequently cited studies are often simply shown in poorly labeled graphs or histograms and is not accessible anywhere else. Access to these older datasets could potentially provide information helpful for answering enduring questions related to *n*-alkanes.


Finally, there is a noticeable lack of available data from many areas worldwide, as seen in Figure 7. Most sites with reported *n*-alkane data are concentrated in Europe, though there is a trend in recent studies being performed in less-represented areas, such as Peru (Wu et al., 2019), Ethiopia (Lemma et al., 2019), and Mongolia (Struck et al., 2020). Future studies of *n*-alkanes should similarly aim to cover less researched geographic areas.

## 5.3 Recommendations

Meta-analysis and synthesis have become more prevalent across scientific disciplines, including in the natural sciences, and have enabled better understanding of scientific questions on global scales (Gurevitch et al., 2018). This review provides a rudimentary synthesis of the current state of knowledge on the degradation and transformation of plant-derived *n*-alkanes in soil archives. Due to the knowledge gaps and issues of data accessibility and compatibility, a truly rigorous systematic review

and meta-analysis is currently not possible for biomarker studies. However, if the research community begins to move towards more uniform and systematic reporting of biomarker data, the potential for synthesis across studies could increase rapidly and enable a more complete and quantitative understanding of the fate of *n*-alkanes in soils across ecosystems.



To move towards this goal, we provide three concrete suggestions for anyone who plans to collect plant and soil biomarker

data:

1. Measure and report basic soil parameters, such as bulk OC and soil pH. If the soil type has already been characterized, include a reference to this information. If the soil has not been characterized, the soil type should be determined using the World Reference Base for Soil Resources (IUSS Working Group, 2014) or the USDA Soil Taxonomy (Soil Survey Staff, 2014) guidelines rather than a localized classification system in the interest of
standardization.
2. Local climate information should be reported, or enough specific location data should be provided so that it is easy for readers to find the climate information from other sources. Additionally, local vegetation should be characterized at a minimum by reporting dominant species cover.
3. Longer chain lengths of $n$-alkanes should be measured and reported, e.g., $C_{20}$-$C_{36}$. The primary data is more useful
to report than index measurements due to the variations in equations for index measurements.

In the future, these data should be gathered in a freely accessible database dedicated to lipid biomarker measurements, including $n$-alkanes, that could allow scientists globally to have a better understanding of what data is already available and what study areas are underrepresented. To be built-up, such a database would need a general template of required data including
soil characteristics and environmental parameters in addition to the biomarker concentration and pattern data. In addition, meta-data standards for describing these data would need to be selected (for an overview, see Hoffmann et al. 2020).

Schädel et al. (2020) recently developed a database for soil incubation studies, the Soil Incubation Database (SIDb), along with guidelines for reporting data from the studies. Due to the related nature of the research, we have adapted their reporting
guidelines as a suggested starting point for standardization of biomarker studies (Table 3). As mentioned, site information including soil classification and vegetation should be considered essential to report. Some soil characteristics are essential, particularly bulk OC, pH, and depth. Others are recommended as they could enable a more quantitative understanding of biomarker degradation or preservation. It is preferable to report primary biomarker data rather than index measurements, though if index measurements are included, the equations used for calculation must also be included. Studies using litterbag
experiments should report some additional data regarding the set-up and execution of the experiment. Wider adoption of these guidelines would be very beneficial for developing a biomarker database. Eventually, such a database could be similar in structure and usability to the Neotoma Paleoecology Database (https://www.neotomadb.org). Until that time, but also as an alternative in case data is incompatible with the structure imposed by the database, scientists could use existing scientific data repositories such as Pangaea (https://www.pangaea.de) or the EarthChem Library (https://www.earthchem.org).

**Data availability.** Data extracted from the reviewed studies may be found in the Supplementary Material.



**Author contribution.** All authors contributed to the conceptualization of this review. CLT performed the data curation, formal analysis, visualization, and preparation of the original draft. CLT and GLBW acquired financial support. All authors

contributed to revision and editing.

**Competing interests.** The authors declare that they have no conflict of interest.

**Acknowledgments.** We gratefully acknowledge funding from the Swiss National Science Foundation under contract 188684

and from swissuniversities in the form of a grant supporting cotutelle de thèse projects. We thank Tatjana Speckert for her

helpful comments on the manuscript during preparation.

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

700





**Figure 1: Percent change in concentration (µg/g dry weight) of *n*-alkanes over time in litterbag experiments.**





**Figure 2: Change in (a) Average Chain Length (ACL) and (b) Carbon Preference Index (CPI) over time in litterbag experiments.**





**Figure 3: (a) Total concentration (µg/g dry weight) of *n*-alkanes in fresh leaves, litter, organic layer (litter removed), and topsoil. For the coniferous forest vegetation type, there were 9 sites included from 1 study (Schäfer et al., 2016). For the deciduous forest type, 39 sites were included from 7 studies (Bliedtner et al., 2018; Bush and McInerney, 2015; Chikaraishi and Naraoka, 2006; Schäfer et al., 2016; Stout, 2020; Trigui et al., 2019; Wu et al., 2019). For the mixed forest type, 25 sites were included from 2 studies (Bush and McInerney, 2015; Howard et al., 2018). For the grass- or shrubland vegetation type, 110 sites were included from 8 studies (Bliedtner et al., 2018; Bush and McInerney, 2015; Howard et al., 2018; Lemma et al., 2019; Li et al., 2018; Schäfer et al., 2016; Trigui et al., 2019; Yao et al., 2019). For the peat vegetation type, 5 sites were included from 2 studies (Ficken et al., 1998; Zhang et al., 2017). For the steppe vegetation type, 57 sites were included from 2 studies (Struck et al., 2020; Yao et al., 2019).**

**(b) Total concentration (µg/g OC) of *n*-alkanes in fresh leaves, litter, organic layer (litter removed), and topsoil. For the coniferous forest vegetation type, 5 sites were included from 2 studies (Hirave et al., 2020; Otto and Simpson, 2005). For the deciduous forest vegetation type, 11 sites were included from 5 studies (Angst et al., 2016; Anokhina et al., 2018; Hirave et al., 2020; Otto and Simpson, 2005; Wu et al., 2019). For the mixed forest type, 1 site was included from 1 study (Hirave et al., 2020). For the grass- or shrubland vegetation type, 2 sites were included from 1 study (Otto and Simpson, 2005).**



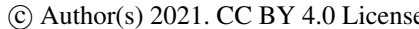



**Figure 4: (a) Average Chain Length (ACL) of *n*-alkanes in fresh leaves, litter, organic layer (litter removed), and topsoil. For the coniferous forest vegetation type, there were 13 sites included from 2 studies (Hirave et al., 2020; Schäfer et al., 2016). For the deciduous forest type, 41 sites were included from 9 studies (Angst et al., 2016; Bliedtner et al., 2018; Bush and McInerney, 2015; Hirave et al., 2020; Lei et al., 2010; Schäfer et al., 2016; Stout, 2020; Trigui et al., 2019; Wu et al., 2019). For the mixed forest type,**

**26 sites were included from 3 studies (Bush and McInerney, 2015; Hirave et al., 2020; Howard et al., 2018). For the grass- or shrubland vegetation type, 110 sites were included from 8 studies (Bliedtner et al., 2018; Bush and McInerney, 2015; Howard et al., 2018; Lemma et al., 2019; Li et al., 2018; Schäfer et al., 2016; Trigui et al., 2019; Yao et al., 2019). For the peat vegetation type, 5 sites were included from 2 studies (Ficken et al., 1998; Zhang et al., 2017). For the steppe vegetation type, 58 sites were included from 3 studies (Buggle et al., 2010; Struck et al., 2020; Yao et al., 2019).**

**(b) Carbon Preference Index (CPI) of *n*-alkanes in fresh leaves, litter, organic layer (litter removed), and topsoil. For the coniferous forest vegetation type, there were 13 sites included from 2 studies (Hirave et al., 2020; Schäfer et al., 2016). For the deciduous forest type, 42 sites were included from 10 studies (Angst et al., 2016; Bliedtner et al., 2018; Bush and McInerney, 2015; Chikaraishi and Naraoka, 2006; Hirave et al., 2020; Lei et al., 2010; Schäfer et al., 2016; Stout, 2020; Trigui et al., 2019; Wu et al., 2019). For the mixed forest type, 26 sites were included from 3 studies (Bush and McInerney, 2015; Hirave et al., 2020; Howard et al., 2018). For**

**the grass- or shrubland vegetation type, 110 sites were included from 8 studies (Bliedtner et al., 2018; Bush and McInerney, 2015; Howard et al., 2018; Lemma et al., 2019; Li et al., 2018; Schäfer et al., 2016; Trigui et al., 2019; Yao et al., 2019). For the peat vegetation type, 5 sites were included from 2 studies (Ficken et al., 1998; Zhang et al., 2017). For the steppe vegetation type, 57 sites were included from 2 studies (Struck et al., 2020; Yao et al., 2019).**

**(c) Odd-over-Even Predominance (OEP) of *n*-alkanes in fresh leaves, litter, and topsoil. For the coniferous forest vegetation type,**

**there were 13 sites included from 2 studies (Hirave et al., 2020; Schäfer et al., 2016). For the deciduous forest type, 44 sites were included from 9 studies (Angst et al., 2016; Anokhina et al., 2018; Bliedtner et al., 2018; Bush and McInerney, 2015; Hirave et al., 2020; Schäfer et al., 2016; Stout, 2020; Trigui et al., 2019; Wu et al., 2019). For the mixed forest type, 26 sites were included from 3 studies (Bush and McInerney, 2015; Hirave et al., 2020; Howard et al., 2018). For the grass- or shrubland vegetation type, 105 sites were included from 7 studies (Bliedtner et al., 2018; Bush and McInerney, 2015; Howard et al., 2018; Lemma et al., 2019; Li et al.,**

**2018; Schäfer et al., 2016; Trigui et al., 2019). For the steppe vegetation type, 52 sites were included from 1 study (Struck et al., 2020).**







**Figure 5: (a) Total concentration (µg/g dry weight) of *n*-alkanes in soil. (b) Total concentration (µg/g OC) of *n*-alkanes in soil.**





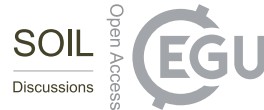

**Figure 6: (a) Average Chain Length (ACL) of *n*-alkanes in soil. (b) Carbon Preference Index (CPI) of *n*-alkanes in soil. (c) Odd-over-Even Predominance (OEP) of *n*-alkanes in soil.**

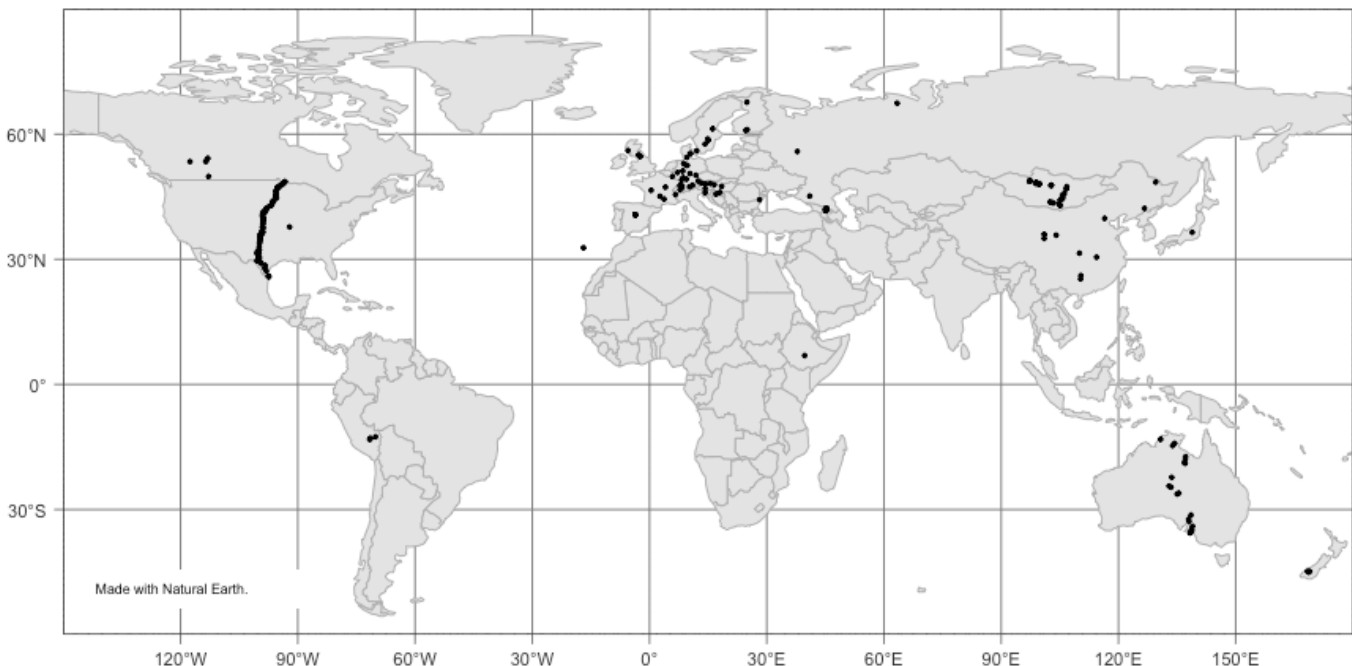

**Figure. 7. Locations of studies included in data analysis.**


**Table 1. Site characteristics for litterbag experiments.**

| Study | Location | Latitude[1] | Longitude[1] | Altitude (m) | MAAT (°C) | MAP (mm) | Soil type | Soil pH | Total duration (days) |
|---|---|---|---|---|---|---|---|---|---|
| Huang et al., 1997 | Moor House Nature Reserve (UK) | 54.688889* | -2.361389* | 550 | 5.1 | 1880 | - | - | 8395 |
| Li et al., 2017 | Wuhan, China | 30.516389* | 114.4025* | - | 16 - 17.5 | 1270 | - | 6.5 – 7.0 | 380 |
| | Guilin, China | 25.281389* | 110.316389* | - | 18.8 | 1874 | - | 5.5 – 6.0 | 369 |
| Nguyen Tu et al., 2017 | "Breuil-Chenue" experimental | 47.302778 | 4.067778 | 650 | 9 | 1280 | Dystric cambisol | 4 - 4.5 | 903 |





| | | | | | | | | | |
|---|---|---|---|---|---|---|---|---|---|
| | forest site (France) | | | | | | | | |
| Schulz et al., 2012 | Incubation experiment | - | - | - | 14 | - | - | 6 | 210 |
| Wang et al., 2014 | Shangzhuang Experimental Station (China) | 39.8* | 116.466667* | 40 | 13 | 480 | - | 8.13 | 180-210 |
| Zech et al., 2011 | Fichtelgebirge (Northeast Bavaria, Germany; | 50.143056 | 11.869444 | 780 | 5 | 1100 | Albic Rustic Podzol | - | 810 |

1. For studies that did not provide exact coordinates, the latitude and longitude were estimated based off of the location description provided. Estimated coordinates are marked with an asterisk.


**Table 2. Site characteristics for open plant-soil systems.**

| Study | Location | Latitude[1] | Longitude[1] | Altitude (m) | MAAT (°C) | MAP (mm) | Soil type | Soil pH | Vegetation |
|---|---|---|---|---|---|---|---|---|---|
| Almendros et al., 1996 | Spain (12 sites) | Between 40.681429 and 40.7017498 | Between -3.668755 and -3.639867 | 640-1200 | - | - | 10 Cambisols, 1 Regosol, 1 Leptosol | 5.4-7.7 | *Pinus pinea, Quercus rotundifolia,* or *Juniperus thurifera* |
| Andersson and Meyers, 2012 | Lek-Vorkuta region, Russia | 67.416666 | 63.3666667 | 160 | -5.80 | 531 | - | - | *Vaccinium vitis-idaea, Ledum palustre* |
| Angst et al., 2016 | Grinderwald, Germany | 52.566667 | 9.3 | - | - | - | Dystric Cambisol | 3.4-4.5 | *Fagus sylvatica* |
| Anokhina et al., 2018 | Losiny Ostrov National Park, Moscow (4 sites) | 55.8663399* | 37.8326505* | - | - | - | Albic Retisol | - | *Tilia cordata, Corylus avellana, Acer platanoides, Sorbus aucuparia* |
| Bliedtner et al., 2018 | Eastern Georgia (22 sites) | Between 41.71030 and 42.26185 | Between 45.055283 and 45.359400 | 445-1659 | 5.4 - 13.3 | 600 - 2000 | - | - | Grassland or deciduous forest |





| | | | | | | | | | dominated by hornbeam |
|---|---|---|---|---|---|---|---|---|---|
| Buggle et al., 2010 | Mircea Voda, Romania | 44.316667 | 28.18333 | - | - | - | Calcic Chernozem | - | Steppe |
| Bush and McInerney, 2015 | Transect through middle of United States (70 sites) | Between 25.850433 and 48.5989 | Between -93.233217 and -100.5171 | - | 3.3=23.4 | 559-736 | | - | mixed temperate forest, tall grass and mixed grass prairies, savanna and sub-tropical forest |
| Celerier et al., 2009 | Mignaloux-Beauvoir, France | 46.5569367* | 0.41096902* | - | - | - | Luvic cambisol | 6.4-7.7 | Grassland |
| Chikaraishi and Naraoka, 2006 | Lake Haruna, Japan | 36.4666667 | 138.866667 | | | | | | *Acer argutum, Acer carpinifolium* |
| Feng and Simpson, 2007, Otto et al., 2005 | Lethbridge, Alberta, Canada | 49.865942* | -112.82635* | - | - | 413 | Brown chernozem | 6-6.9 | Grassland |
| | | | | | | - | Dark brown chernozem | | |
| | Edmonton, Alberta, Canada | 53.567702* | -113.49551* | - | - | 452 | Black chernozem | - | |
| | University of Alberta Ellerslie Research Station | 53.440783* | -113.54699* | - | - | - | Eluviated black chernozem | - | |
| Ficken et al., 1998 | Moine Mhor, Scotland | 56.0886535447229* | -5.509956860782824* | 950 | 5 | - | - | 4 | *Eriophorum vaginatum, Carex bigelowii, Racomitrium lanuginosum.* |





| | | | | | | | | | |
|---|---|---|---|---|---|---|---|---|---|
| Hirave et al., 2020 | Southern Black Forest (Germany) | 47.661944 | 7.782778 | 535 | 8.9 | 1155 | Stagnosol | 3-4.2 | *Fagus sylvatica* |
| | | 47.864167 | 8.102778 | 965 | 3.9 | 1800 | Podzol | | *Picea abies* |
| | | | | | | | | | *Sphagnum quinquefarium* |
| | Lake Baldegg catchment (Switzerland) | 47.168056 | 8.259167 | 608 | 8.4 | 1100 | Cambisol | | *Picea abies, Thuidium tamariscinum* |
| | | 47.166389 | 8.262500 | 573 | 8.4 | 1100 | Cambisol | | *Picea abies* |
| | Upper Sûre Lake catchment (Luxembourg) | 49.865833 | 5.857500 | 429 | 9.2 | 970 | Cambisol | | *Picea abies, Quercus robur* |
| Howard et al., 2018 | 20 sites on a north-south transect across Australia | Between -35.608 and -13.158 | Between 130.78 and 138.96 | - | Between 14.03 and 27.44 | Between 194.65 and 1642.9 | - | - | Various |
| Huang et al., 1996 | Moor House Nature Reserve, UK | 54.71909* | -2.462827* | 750-775 | - | - | Peaty gley | < 4.5 | |
| | | | | | | | Acid brown earth | | |
| | | | | | | | Podzol | | |
| Lehtonen and Ketola, 1993 | Kaurastensuo bog (Lammi, southern Finland) | 61.0775* | 25.011* | - | - | - | - | - | *Sphagnum* |
| | Norrbomuren fen (Gävleborg county, eastern Sweden) | 61.3012* | 16.1534* | | | | | | *Sphagnum, Carex* |

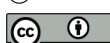



| | Ahvenjärven vuoma mire (Kittilä, northern Finland) | 67.6507* | 24.9158* | | | | | | *Bryales* |
|---|---|---|---|---|---|---|---|---|---|
| | Suurisuo mire (Janakkala, southern Finland) | 60.9201* | 24.646* | | | | | | *Carex, Bryales* |
| Lei et al., 2010 | Xinglong Mountain, China | 35.78333 | 104.066667 | 2600 | 4.10 | 625.00 | - | - | Populus davidiana and Betula platyphyla |
| Lemma et al., 2019 | Bale Mountains, Ethiopia (27 sampling sites) | 6.9241* | 39.7024* | 2550 to 4377 | - | - | - | - | Afroalpine scrub, heath, montane forests |
| Li et al., 2018 | Mt. Cardrona, New Zealand | -44.85* | 168.95* | 400-1300 | 9-11 | 600-900 | Typic Haplorthods | - | Grassland |
| | Rees Valley, NZ | -44.80320623* | 168.39690910* | | | | | | |
| | Queenstown, NZ | -45.0302* | 168.6616* | | | | | | |
| Marseille et al., 1999 | south slope of Mont-Lozère, France | 44.42611* | 3.73917* | 1300-1500 | - | - | Ranker and Dystric Podzoluvisol | 5.2 - 5.4 | *Fagus sylvatica, Vaccinium myrtillus, Calluna vulgaris* |
| | | | | | | | | 4.9-5.6 | *Picea abies, Sorbus aucuparia, Deschampsia flexuosa,* |

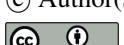



| | | | | | | | | 4.4-4.6 | *Festuca ovina, Festuca rubra, Nardus stricta, Cytisus purgans* |
|---|---|---|---|---|---|---|---|---|---|
| | | | | | | | | | *Oxalis acetosella* |
| Naafs et al., 2004 | Madeira Island (Portugal) | 32.759524* | -16.895778* | 1175 | - | - | Andisol | 4-4.5 | Grass |
| Nguyen Tu et al., 2001 | Massif Central, France | 45.1093* | 2.6753* | - | - | - | - | - | *Gingko biloba* |
| Otto and Simpson., 2005 | Lethbridge, Alberta, Canada | 49.865942* | -112.82635* | - | - | - | Brown Chernozem / Dark Brown Chernozem | 6-6.9 | Western Wheatgrass (*Agropyron smithii*) |
| | Strathcona County, Alberta, Canada | 54.211654* | -113.02797* | - | - | - | Dark Gray Chernozem | - | Quaking Aspen (*Populus tremula*) |
| | Hinton, Alberta, Canada | 53.421871* | -117.57637* | - | - | - | Eutric Brunisol | - | Lodgepole Pine (*Pinus contorta*) |
| Schafer et al., 2016 | 26 locations across Central Europe | Between 45.6515 and 58.87 | Between 7.17 and 21.24 | 16 – 899 | 5.3-11 | 470 - 1742 | - | - | Grass, deciduous, coniferous |
| Stout, 2020 | Missouri | 37.75 | -92.1 | 342 | 12.5 | 1071 | - | - | *Quercus, Acer, Carya* |
| Struck et al., 2020 | Two transects in Mongolia | Between 42.85992 and 48.80367 | Between 97.22267 and 106.7665 | 1224-2792 | -7.3-5.5 | 99.3-276.2 | - | - | *Poaceae, Cyperaceae, Artemisia spp., Caragana spp.* and *Larix sp.* |
| Trigui et al., 2019 | Armenia | 45.166667 | 41 | 680-960 | 11 | 450-550 | - | - | Various |




| van der Voort et al., 2017 | Lausanne | 46.56666667 | 6.65 | 800-814 | 7.6 | 1134 | Cambisol | 4.6 | *Fagus sylvatica* |
|---|---|---|---|---|---|---|---|---|---|
| | Beatenberg | 46.7 | 7.766666667 | 1490-1532 | 4.7 | 1163 | Podzol | 4.1 | *Picea abies* |
| Wu 2019 | Wayqecha, Madre de Dios, Peru | -13.1926 | -71.588 | 3025 | 11.1 | 1560 - 5300 | Umbrisol | - | *Weinmannia crassifolia, Clusia alata cf.,* and *Hesperomeles ferruginea* |
| | San Pedro | -13.049 | -71.537 | 1500 | 18.8 | | Cambisol | - | *Alchornea latifolia, Tachigali setifera,* and *Tapirira obtuse* |
| | Villa Carmen | -12.8961 | -71.4183 | 614 | 22.9 | | - | - | Bamboo |
| | Los Amigos | -12.5588 | -70.0993 | 286 | 24.4 | | Ultisol | - | *Inga* |
| Xie et al., 2004 | Bolton Fell Moss, Cumbria, England | 55.011389* | -2.798056* | - | - | - | - | - | *Sphagnum* mosses |
| Yao et al., 2019 | Mountain Laji, China | 36 | 101 | 2717-4200 | 7.20 | 254.00 | - | - | *Potentilla fruticosa, Stipa purpurea,* |
| | Mountain Anizhihai, China | 35 | 101 | 2622-4157 | | | - | - | *Carex dispalata, Kobresia pygmaea, Kobresia humilis, Elymus nutans, polygonum vivparum, Polygonum sibiricum, Festuca ovina,* |


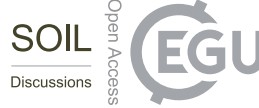

| | | | | | | | | | Caragana sinica, Taraxacum mongolicum, Leontopodium nanum, and Poa pratensis. |
|---|---|---|---|---|---|---|---|---|---|
| Zhang et al., 2017 | Yichun peatland, China | 48.516667 | 129.533333 | 370 | - | - | - | - | Sphagnum recurvum, Sphagnum palustre, Sphagnum jensenii, Sphagnum magellanicum, Sphagnum cuspidatulum |
| | Hani peatland, China | 42.216667 | 126.651667 | 900 | 4 | 757–930 | - | - | Sphagnum multifibrosum, Sphagnum acutifolioides |
| | Dajiuhu peatland, China | 31.466666 | 110 | 1700 | 7.2 | 1560 | - | - | Sphagnum palustre, Sphagnum compactum |
| | Shiwangutian peatland, China | 26.083333 | 110.366667 | 1690 | 12-13 | 1800 | - | - | Sphagnum magellanicum, Sphagnum multifibrosum, Sphagnum palustre, Sphagnum portoricense |

1. For studies that did not provide exact coordinates, the latitude and longitude were estimated based on the location description provided. Estimated coordinates are marked with an asterisk.






**Table 3. Data reporting guidelines adapted from Schädel et al. (2020)**

| Variable | Suggested units | Essential/ Recommended | Notes |
|---|---|---|---|
| *Site information* | | | |
| Latitude/longitude | Decimal degrees | E | |
| Mean annual air temperature | °C | E | |
| Mean annual precipitation | mm | E | |
| Soil classification | | E | Use the WRB or USDA guidelines |
| Vegetation | | E | |
| *Soil characteristics* | | | |
| Soil depth | m | E | |
| Soil horizon | | R | |
| SOC | mg C/ g dry weight or % | E | |
| pH | | E | |
| Texture | % clay, silt, sand | R | |
| *For plant samples* | | | |
| Species | | E | |
| Season of sampling | | R | |
| *Biomarker data* | | | |
| Concentration of individual biomarkers | µg/g dry weight or µg/g OC | E | |
| Index measurements (e.g., ACL, CPI, OEP) | | R | |
| Formula(s) used for index measurements | | R | |
| *Extra variables for litterbag experiments* | | | |
| Duration | days | E | |
| Depth | m | E | If litterbags were buried |
