# Peer review of "Transformation of *n*-alkanes from plant to soil: a review"

_SOIL, 2020_

## Author Response (AR1)

***Response to Referee #1***

We would first like to thank Anonymous Referee #1 for their critical review and helpful comments and suggestions for our manuscript.

**Specific Comments**

***Anonymous Referee #1:*** *Write impersonal, mainly when this is a review paper*

**Reply:** Thank you for the suggestion. We will make adjustments in the revised version.

***Anonymous Referee #1:*** *Check and Italicize the "n" for "normal" e.g. "n-alkanes" in the Abstract section*

**Reply**: Thank you for catching this mistake. We will correct this.

***Anonymous Referee #1:*** *Paragraph 35. Apart from Kolattukudy et al., 1976, I think that the pioneering works of Eglinton et al., 1961a and b should be cited:*

*Eglinton et al (1962a) Nature DOI: 10.1038 / 193739a0*

*Eglinton et al (1962b) Phytochemistry DOI: 10.1016 / S0031-9422 (00) 88006-1*

**Reply**: Thank you for the suggestion of these citations. We agree and will include them in the revised version.

***Anonymous Referee #1:*** *Paragraphs 55-65. Please, check the formulas (CPI, OEP & ACL) and normalize notation i.e. all in summation using the appropriate indexes (n, m)*

**Reply**: Thank you for the suggestion. We will check the formulas and notation for the revised version.

***Anonymous Referee #1:*** *Paragraphs 65-70. Here I missed some other relevant and general citations related to n-alkanes and other biomarkers diagenesis in soils and sediments.*

*Bourbonniere & Meyers (1996). DOI: 10.4319 / lo.1996.41.2.0352*

*Wiesenberg et al (2003). DOI: 10.1016 / j.orggeochem.2004.03.009*

*Meyers & Ishiwatari (1993). DOI: 10.1016 / 0146-6380 (93) 90100-P*

*Zhang et al (2006). DOI: 10.1016 / j.quascirev.2005.03.009*

**Reply**: Thank you for the suggestion of these citations. We will include them in the revised version.

***Anonymous Referee #1:*** *In addition, although the following references are cited in other parts of the MS, I will also recommend its inclusion in this introductory paragraph.*

*Bull et al (2000). DOI: 10.1016 / S0146-6380 (00) 00008-5*

*Otto & Simpson (2005). DOI: 10.1007 / s10533-004-5834-8*

**Reply**: Thank you for this suggestion. We will also cite these in the introduction.

***Anonymous Referee #1:*** *Paragraph 330. Maybe it is worth* to *briefly mention other environmental aspects known to exert shifts in n-alkanes e.g. forest fires.*

*Almendros et al (1988). DOI: 10.1016 / 0016-7061 (88) 90028-6*

*González-Pérez et al (2008). DOI: 10.1016 / j.orggeochem.2008.03.014*

**Reply**: Thank you for the suggestion and the suggested citations. We agree that it is important to consider these other potential causes for changes in n-alkanes and will include this in the revised version.

***Relevant changes:***
- We italicized all "*n*-alkanes" where previously missing.
- We added suggested citations.
- We homogenized the equations as suggested.

***Response to Referee #2:***

We would first like to thank Anonymous Referee #2 for their critical review and helpful comments.

***Anonymous Referee #2:*** *My main comment concerns the soil profiles. The authors described the trends, however for many results the data available are from surface to around ten centimeter. In many soils, these ten centimeters often only concern the organic layer. Accordingly, it should be clearly stated as it could be an important point. For example in figure 5a, we can observe important difference in n-alkane concentration in the first 20 cm and below. The distinction was better evidenced in figures 3 and 4. With the differences of scale, it is sometimes difficult to compare the evolution with depth (Figures 5 and 6). Perhaps it might be interesting to distinguish the trend between 0-20 cm and between 20 to deep soil in the discussion.*

**Reply:** We agree that for many of the longer soil profiles there is a difference in the trend between the upper and lower depths. For many of the profiles, there was only data available for the organic layer which as you mention is an important point to consider. We will revise our discussion in the future version to more fully address the effects of depth.

***Anonymous Referee #2:*** *In the introduction, why do the authors specified that the study of lipids specifically could increase the overall understanding of SOM dynamics? Would it signify that the authors consider that all individual compounds have the same degradation and preservation pattern? However the authors mentioned potential preservation of n-alkanes. So the authors should improve how they would apply knowledge on alkane degradation on the whole SOM if they are better preserved.*

**Reply:**

Whether or not certain molecular component classes have a larger or smaller potential for preservation as compared to bulk SOM is currently a matter of scientific debate. While the currently prevailing paradigm has it that there is no intrinsic recalcitrance of certain molecular classes (e.g. Schmidt et al., Nature, 2011; Lehmann & Kleber, Nature, 2015), insights are emerging that SOM turnover rates are linked to functional complexity with an important role for variations in molecular diversity of SOM (Lehmann et al., Nature Geoscience,2020). Lipids constitute and important, and molecularly very diverse sub-class of SOM and as such studying lipid dynamics in soils under various pedogenic and environmental conditions will help further the debate on SOM dynamics.

We will include this clarification in the revised version of the introduction.

**Specific Comments**

***Anonymous Referee #2:*** *L104: Replace where by when*

**Reply**: Thank you for the correction. We will fix in the revised version.

***Anonymous Referee #2:*** *L 155 and L169 to 172: Are the trends noticed really significant statistically? In fig 3a perhaps it is significant for coniferous forest, mixed forest, and in fig 3b too, except deciduous but for fig 4 it is even more difficult to know if the differences are significant. Could it be possible to apply statistical analysis?*

**Reply:** In line 155, "significantly" is used in a colloquial rather than statistical sense. Although we contemplated completing a statistical analysis on the compiled data, we chose not to because we do not want to risk overinterpreting the data. Due to the limited size of the dataset that could be gathered, as well as differences across studies, it would be difficult to determine what populations are accurately represented in the data.  We will remove the colloquial use of the word 'significant' in the revised version of the manuscript.

***Anonymous Referee #2:*** *L226-230: I totally agree with the authors about the limitation. We need data to evaluate trends. Thus, the "trajectory plant-litter-soil" mentioned is perhaps too large because for coniferous forest and grassland the data are from very shallow layer (fig 5a), perhaps even only litter. It is difficult to write that it is a trend from plant to soil.*

**Reply:** We agree that describing our observed trends here as covering the entire trajectory from plant to soil is too optimistic considering the limitations of the data. Therefore, we will be more specific in the revised version.

***Anonymous Referee #2:*** *L238: I do not remember that Jansen and Nierop (2009) discussed the production of alkane via alkene oxidation. The authors quote another citation for this possible source of alkanes.*

**Reply:** Thank you for pointing out this error. We will adjust the citations in the revised version.

***Anonymous Referee #2:*** *L245-247: I do not understand how n-alkane degradation could result in an increase of n-alkane concentration.*

**Reply:** In these lines, we tried to briefly summarize how Wang et al. (2014) explained the unexpected results of their study. They did not say that n-alkane degradation would cause an increase in concentration, but that the process of degradation is complex due to multiple factors and that the unexpected increase in n-alkane concentration in the litterbags in a short-term experiment was evidence for this complexity. Thank you for pointing out that this was not well-phrased in the current version. We will clarify this in the revised version.

***Anonymous Referee #2:*** *L290-291: How could earthworm influence the dis2009 should be Zech et al., 2010.*

**Reply:** Unfortunately, this comment is not entirely clear, but we would be very happy to address it if you are able to elaborate.

***Relevant changes:***
- We separated the data for the soil profiles into 0-20 cm and 20 cm and below to better analyze the effects of depth.
- We clarified the suggested parts of the introduction and discussion.
- We removed an incorrect citation.

***Response to Referee #3:***

First of all, we thank Anonymous Referee #3 for the preparation of his/her critical review and appreciate receiving some comments that will certainly help to improve the manuscript. However, we do not fully agree with all of the reviewer's statements and opinions, which also frequently contradict the assessments provided by the other two anonymous referees. Overall, we try to follow all advice given by the Referee as much as possible. However, we can neither change the data nor the general observations. We are grateful that we received two additional reviews, which were, to our opinion, well-founded and more constructive than the review performed by Anonymous Referee #3, who seemed to have overlooked some of our key messages and the general difficulties that arise from extracting data from multiple, quite diverse data sources. We hope that with the responses given to Anonymous Referee #3 and the anticipated changes we can provide an improved version of our manuscript.

***Anonymous Referee #3****: "With 'transformation of n-alkanes from plant to soil' Thomas and co-authors have chosen a topic for their review that is an extremely narrow field of research. This is acknowledged by the authors already in their abstract stating 'only a limited number (of studies) have focused on the transformation of these compounds… in soil archives'. Moreover, there is according to my knowledge no discussion or controversy in the scientific community concerning* transformation *of alkanes from plants to soils. This likely explains why no questions are raised by the authors* in *or at the end of the introduction chapter. I* therefore *doubt that the chosen topic merits a review paper that shall attract attention and address a broader readership."*

**Reply**: We do not agree with the statement that the transformation of n-alkanes from plant to soil is a narrow field of research and that this review manuscript cannot attract attention of a broader readership. The search strings of the systematic literature search resulted in 9297 results dealing with a related topic, i.e., lipid biomarkers and mostly alkanes in soils. As alkanes are often better preserved in soils than other compounds, alkanes have been frequently used for source apportionment of plant-derived organic matter in soils. As several publications showed, even alkane composition changes with degradation in plant-soil systems. However, a systematic assessment of the transformation of alkane composition and underlying degradation processes to our knowledge has not been published before.

Although there is no general controversy in the literature on this topic, the generalization of the observations and focus on potential differences, e.g., between different ecosystems, biomes or soil types, is necessary to better understand the fate of alkanes in plant-soil systems. The other reviewers appreciated seeing the data being summarized in the review manuscript (Anonymous Referee #1: "The review is pertinent and appropriately compiles the main findings described in the most relevant publications dealing with alkane biomarker distribution in soils. To the best of my knowledge, the review is novel and not previously published."; Anonymous Referee #2: "This review is interesting despite the small dataset selected due to the limited number of suitable papers. … The manuscript depicts most of the outcome explaining the evolution of n-alkane pattern with time or in soil either due to degradation pathway or source shift.").

The large number of article views since online publication of the preprint of this manuscript indicates that there is a large interest in the topic with a broad readership of SOIL being interested. In fact, the numbers in the article metrics on SOILD show that there are more article views for our preprint of the manuscript than for any of the other individual preprint manuscripts published over the last weeks.

***Anonymous Referee #3***: *Moreover, the readers of 'SOIL' do not learn anything new and the manuscript contains flaws. The 'major findings' summed up by the authors (decreasing n-alkane concentrations and decreasing CPI) are trivial, known for a long time* and *described by more than 90% of the cited respective studies.*

**Reply**: Although it seems "trivial" or well-known that n-alkane concentrations and CPI decrease with degradation, there was no general information available on the order of magnitude so far and if this is identical in all ecosystems and biomes. However, such quantitative information is essential to be able to interpret alkane composition shifts in soils in a systematic way, for instance for the purpose of the reconstruction of past vegetation patterns. Therefore, we summarized the available information rather than relying on fragmented information in different studies.

***Anonymous Referee #3***: *The first part of the third 'major finding' (preferential degradation of odd chain length) is equal to* major *finding (2) just in other words and the second part of the third 'major finding' (preferential degradation of* shorten *chain length n-alkanes) is simply wrong and not supported by the majority of the studies cited by the authors (see ll. 164ff and l. 262).*

**Reply**: We thank the reviewer for this comment. We will combine the major findings 2 and 3 as they are rather similar and do not deserve separate numbering. However, we think that the Referee misunderstood our statement in lines 164ff "Other studies noted decreases in the relative concentration of long-chain n-alkanes (Chikaraishi and Naraoka, 2006; Otto and Simpson, 2005; Hirave et al., 2020), while Nguyen Tu et al. (2001) noted a preferential decrease in shorter chain lengths from fresh leaves to litter of *Gingko biloba*." Most of the authors found either a decrease in shorter chain lengths or an increasing relative concentration of long-chain n-alkanes, which is identical to a relative depletion of short-chain alkanes when compared to long-chain alkanes. In Fig. 2 and 4 this leads to increasing ACL values in most of the studies from plant material towards mineral soil. We will modify the text to "…increase in concentration…".

***Anonymous Referee #3***: *Actually interesting or striking features such as the accumulation of soil microbial-derived medium-chain n-alkanes or the increase of n-alkane concentrations at coniferous forest sites (Fig. 3b) are unfortunately not or insufficiently emphasized or wrongly explained (the increase can be simply explained with needles producing no n-alkanes but understory in coniferous forests contributing to the soil n-alkane pool).*

**Reply**: Unfortunately, there is scarce literature available on microbial sources of alkanes in soils and the available literature is quite old and was not confirmed by newer studies to the best of our knowledge. All of our ACL calculations for studies from which primary data were available were based on data ranging from 27 to 33 carbons (line 100). Thus, mid-chain alkanes (typically with chain length of 20 to 25) are not entirely included and short-chain alkanes (<20 carbons) were entirely excluded, preventing us from drawing conclusions on these components. Thus, the connection of the data to microbial-derived mid-chain alkanes is not possible. We agree that making such a connection is valuable and will emphasize in our conclusions the direction of future study needed for this, now that our review has shown that it is not possible based on the presently available data.

***Anonymous Referee #3***: *A review focussing on* plant *to soil transformation should not include subsoils or peat archives. Statements or citations like in l. 200 or alkane depth functions of peat archives like in Fig. 5 are not helpful and in the* worst case *misleading, because in steppe biomes there is high bioturbation in typically loose eolian sediments and in peat archives the vegetation may have changed.*

**Reply**: We disagree with the referee as: (1) incorporation of plant-derived alkanes is not limited to the top of the soils and litterfall of aboveground biomass, but can include contributions of soil alkanes from roots, which is much stronger in deeper part of the soils than in the top layer. (2) Transformation of organic matter is continuing in deeper soil layers, if litter-derived alkanes are translocated, e.g., by particulate transport. Therefore, only the whole continuum from fresh plant leaves, which is often taken as the sole source of soil alkanes, towards deeper soil horizons can reflect all transformation processes. Strong bioturbation, indeed occurs in steppe biomes, but is not limited to these biomes. Bioturbation will of course influence the vertical stratigraphy of n-alkanes. However, that does not mean one cannot interpret n-alkane patterns with depth. Where mixing via bioturbation may complicate the interpretation of n-alkane patterns for certain purposes such as paleo-ecological reconstructions, it may in fact enhance their applicability for other purposes, e.g. reconstructing SOC transformations as influenced

by bioturbation. Of course, peat archives are special types of hydromorphic soils, where degradation of organic matter is limited. Nevertheless, they are part of the whole soil domain and thus reflecting a considerable part of the (hydromorphic) soils worldwide. Similar to oxic soils, it is important to understand the transformation processes of alkane in hydromorphic soils. This is particularly so, because the biomass accumulation in peat deposits leads to a favorable time/depth axis that makes them valuable archives of n-alkanes for paleo-ecological reconstructions (e.g. Jansen et al.,2010: https://doi.org/10.1016/j.palaeo.2009.10.029) Therefore, and as both of the other referees did not raise such concerns, we prefer to keep the whole sample set inside the manuscript.

***Anonymous Referee #3****: Apart from Fig. 5, also Figs. 3, 4* and *6 are hardly readable. Concerning Fig. 3b, I can hardly imagine (actually it cannot be) that fresh deciduous forest material and* fresh *mixed forest material* contains *no alkanes. Please check and correct your data and figures.*

**Reply**: Many thanks for mentioning the readability. We shall try to improve this by increasing the font size of the figures. In Figure 3b, the figure does not show that there are no alkanes in the fresh deciduous forest material and fresh mixed forest material, though the values are quite low. As is noted in the caption, Figure 3b shows the total concentration of n-alkanes relative to the amount of organic carbon. Therefore, due to high amounts of organic carbon in fresh material, the relative concentration of n-alkanes is quite low. In the revised version, we will try to further improve the readability of these figures by adjusting the sizes of the axis so that it is more clear that these values are not at zero.

***Anonymous Referee #3****: Last but not least, it does not become clear what the knowledge gaps are. The authors encourage expanding the dataset to less researched geographic areas… I consider it to be rather unlikely that this approach will help* increasing *our understanding of plant to soil transformation of n-alkanes.*

**Reply**: There are multiple knowledge gaps that are not limited to geographic areas. The major issue that was highlighted in the review was the limited comparability of the data which coincides with the diverse reporting of data and even the lack of additional information that is published together with molecular data, which is why we came to our recommendations. With increasing need to collectively analyze all available data like Big Data analytics, it became obvious that even molecular data needs to be reported according to the FAIR principles to better use the data also in future research.

***Anonymous Referee #3****: To sum up, the issues raised above demonstrate that the overall aim formulated by the authors at the end of the introduction (l. 68ff: 'consolidation of the available information on the fate of n-alkanes in soils… better process understanding of degradation…') is only inadequately achieved. Most importantly, soil microbial build-up of n-alkanes is insufficiently addressed.*

**Reply**: We disagree with the referee in these points, which also contradict the conclusions of the other two referees. For instance Anonymous Referee #1 wrote "The review is pertinent and appropriately compiles the main findings described in the most relevant publications dealing with alkane biomarker distribution in soils.", which contradicts the statement that we "inadequately achieved" the "overall aim". We were

overwhelmed to find almost 9300 articles after our systematic literature search, but the disappointment was that after half a year of screening these articles only 37 articles contained enough relevant and extractable information that allowed us to properly summarize the data. The reasons were quite diverse but clearly show that although there have been many studies performed, it is extremely hard to extract this data and to make sense out of this. We hope our manuscript can help to improve these issues in the future. As mentioned before, microbial alkanes were not the focus of the review and most of the studies that we found did not include data on mid- or short-chain lengths as we would have included also these in the data evaluation otherwise.

**Specific Comments**

*Anonymous Referee #3*: *l. 48 and 50: I exemplarily checked both Marzi et al., 1993 and Hoefs et al., 2002 and found them to be inappropriately cited. Marzi and Hoefs use CPI and OEP, but not in the sense that their studies or results support what the authors cite them for, namely well preserved or highly degraded plant organic matter. Please be more specific with your citations.*

**Reply**: Many thanks for this comment. We will replace these with Cranwell,1981, Organic Geochemistry and Zech et al., 2009, E&G Quaternary Science Journal.

*Anonymous Referee #3*: *Result chapter: numbering of subchapter makes no sense*

**Reply**: Many thanks for pointing to this. We will modify the subchapters to 3.1, 3.2, and 3.3 rather than 3.1.1, 3.1.2, and 3.1.3. and we will remove the current headline 3.1. This was a leftover of a previous version of the manuscript.

*Anonymous Referee #3*: *l. 238: cannot be correct, oxidation of alcohols does certainly not produce n-alkanes. The succession of oxidation is aliphatic – aldehyde – alcohol – acid.*

**Reply**: Thanks a lot for this comment. We will modify this in the revised version.

*Anonymous Referee #3:* *l. 283ff: I do not agree with the statement that 'retaining the range of chain length and the most abundant chain length' 'evidences that there is limited change… no preferential degradation...'. Fig. 2a shows that all ACL lines increase.*

**Reply**: We kindly point the reviewer to the supplement as indicated in line 284, where it becomes more obvious that there is no preferential degradation or preservation that can be drawn. The range of alkanes as well as the most abundant compound always stays the same. The changes in the ACL in Fig. 2a are rather small. However, we will rephrase that part.

*Relevant changes:*
- We corrected incorrect citations.
- We removed an incorrect reference.
- We fixed the numbering of the results chapter.
- We increased the sizes of the figures.
- We combined two of our original findings.

***Additional changes:***
- We homogenized the references.

---

## Author Response (AR2)

Author's response

- Added citation as suggested by editor
- Fixed minor typos